# Atrial fibrillation and comorbidities: Clinical characteristics and antithrombotic treatment in GLORIA-AF

**Monika Kozieł**[1,2], **Christine Teutsch**[3], **Jonathan L. Halperin**[4], **Kenneth J. Rothman**[5], **Hans-Christoph Diener**[6], **Chang-Sheng Ma**[7], **Sabrina Marler**[8], **Shihai Lu**[8], **Venkatesh K. Gurusamy**[9], **Menno V. Huisman**[10], **Gregory Y. H. Lip**[1,2,11] *, on behalf of the GLORIA-AF Investigators**¶**

1 Liverpool Centre for Cardiovascular Science, University of Liverpool and Liverpool Heart & Chest Hospital, Liverpool, United Kingdom, 2 1st Department of Cardiology and Angiology, Silesian Centre for Heart Diseases, Zabrze, Poland, 3 Department of Clinical Development and Medical Affairs, Therapeutic Area Cardiometabolism, Boehringer Ingelheim International GmbH, Ingelheim, Germany, 4 Icahn School of Medicine at Mount Sinai, New York, New York, United States of America, 5 RTI Health Solutions, Research Triangle Park, North Carolina, United States of America, 6 Institute for Medical Informatics, Biometry and Epidemiology, University of Duisburg-Essen, Essen, Germany, 7 Cardiology Department, Atrial Fibrillation Center, Beijing AnZhen Hospital, Capital Medical University, Beijing, China, 8 Biostatistics and Data Sciences, Boehringer Ingelheim Pharmaceuticals, Inc., Ridgefield, Connecticut, United States of America, 9 Global Epidemiology, Boehringer Ingelheim International GmbH, Ingelheim, Germany, 10 Department of Thrombosis and Hemostasis, Leiden University Medical Center, Leiden, the Netherlands, 11 Aalborg Thrombosis Research Unit, Department of Clinical Medicine, Aalborg University, Aalborg, Denmark

¶ Membership of the GLORIA-AF Investigators Group is listed in the Acknowledgments.
* gregory.lip@liverpool.ac.uk

**Data Availability Statement:** All relevant data are within the manuscript and its Supporting information files.

## Abstract

### Background

Patients with AF often have multimorbidity (the presence of ≥2 concomitant chronic conditions).

### Objective

To describe baseline characteristics, patterns of antithrombotic therapy, and factors associated with oral anticoagulant (OAC) prescription in patients with AF and ≥2 concomitant, chronic, comorbid conditions.

### Methods

Phase III of the GLORIA-AF Registry enrolled consecutive patients from January 2014 through December 2016 with recently diagnosed AF and $CHA_2DS_2$-VASc score ≥1 to assess the safety and effectiveness of antithrombotic treatment.

### Results

Of 21,241 eligible patients, 15,119 (71.2%) had ≥2 concomitant, chronic, comorbid conditions. The proportions of patients with multimorbidity receiving non-vitamin K antagonist oral anticoagulants (NOACs) and vitamin K antagonists (VKA) were 60.2% and 23.6%,

**Funding:** The work was supported by Boehringer Ingelheim, Germany. The funder provided support in the form of salaries for authors [CT, SL, SM, VKG, JH, CD, CM, MH, GYHL], but did not have any additional role in the study design, data collection and analysis, decision to publish, or preparation of the manuscript. The specific roles of these authors are articulated in the 'author contributions' section.

**Competing interests:** The authors have read the journal's policy and have the following competing interests: Dr. Teutsch, Sabrina Marler, and Venkatesh K. Gurusamy are paid employees of Boehringer Ingelheim. Dr. Lu was a paid employee of Boehringer Ingelheim at the time that the manuscript was written. Professor Halperin has engaged in consulting activities for Boehringer Ingelheim and advisory activities involving anticoagulants, and he is a member of the Executive Steering Committee of the GLORIA-AF Registry. Over the past 3 years, Professor Diener received honoraria for participation in clinical trials, contribution to advisory boards, or oral presentations from: Abbott, Bayer Vital, Bristol-Myers Squibb, Boehringer Ingelheim, Daiichi Sankyo, Medtronic, Pfizer, Portola, Sanofi-Aventis, and WebMD Global. Financial support for research projects was provided by Boehringer Ingelheim. He received research grants from the German Research Council (DFG), German Ministry of Education and Research (BMBF), European Union, NIH, Bertelsmann Foundation, and Heinz-Nixdorf Foundation. Professor Ma received honoraria from Bristol-Myers Squibb, Pfizer, Johnson & Johnson, Boehringer Ingelheim, Bayer, and AstraZeneca for giving lectures. Professor Huisman reports grants from ZonMW Dutch Healthcare Fund, grants and personal fees from Boehringer Ingelheim, Pfizer/Bristol-Myers Squibb, Bayer Health Care, Aspen, Daiichi Sankyo, outside the submitted work. Professor Lip has been a consultant for Bayer/Janssen, Bristol-Myers Squibb/Pfizer, Medtronic, Boehringer Ingelheim, Novartis, Verseon, and Daiichi Sankyo. He has been a speaker for Bayer, Bristol-Myers Squibb/Pfizer, Medtronic, Boehringer Ingelheim, and Daiichi Sankyo. No fees directly received personally. These competing interests do not alter our adherence to PLOS ONE policies on sharing data and materials. There are no patents, products in development or marketed products associated with this research to declare. Dr Kozieł and Professor Rothman declare no competing interests.

respectively. The proportion with paroxysmal AF was 57.0% in the NOAC group and 45.4% in the VKA group. Multivariable log-binomial regression analysis found the following factors were associated with no OAC prescription: pattern of AF (paroxysmal, persistent, or permanent), coronary artery disease, myocardial infarction, prior bleeding, smoking status, and region (Asia, North America, or Europe). Factors associated with OAC prescriptions were age, body mass index, renal function, hypertension, history of cerebral ischemic symptoms, and AF ablation.

## Conclusion

Multimorbid AF patients prescribed NOACs have fewer comorbidities than those prescribed VKAs. Age, AF pattern, comorbidities, and renal function are associated with OAC prescription.

## Introduction

Atrial fibrillation (AF) affects approximately 3% of adults and its prevalence and incidence are rising [1] with the aging of the population [2]. Older patients with AF often have other chronic conditions that affect their clinical course [3]. Multimorbidity (the presence of ≥2 concomitant chronic conditions) demands a holistic and integrated approach to patient care [4] since these patients face higher risks of stroke and bleeding than those without comorbidities [5, 6]. The interplay between comorbidity, AF, and optimal thromboprophylaxis has both medical and economic implications [7]

The aim of this analysis of the GLORIA-AF dataset is to describe baseline characteristics and antithrombotic therapy prescription patterns in patients with AF and multimorbidity and to identify factors associated with the selection of an oral anticoagulant (OAC) type for these complex patients.

## Materials and methods

The design of the GLORIA-AF registry (https://clinicaltrials.gov/ct2/home; trial registration numbers NCT01468701, NCT01671007, NCT01937377) has been reported [8]. The study protocol is concordant with the ethical guidelines of the 1975 Declaration of Helsinki, and informed consent was obtained from each patient before enrollment.

The registry collected routine clinical practice data regarding patients with newly diagnosed AF to evaluate patient characteristics influencing the selection, safety, and effectiveness of antithrombotic therapy. Phase I was conducted before non-vitamin K antagonist oral anticoagulants (NOACs) were available for stroke prevention in AF. Phase II began when dabigatran was approved in countries with participating clinical centers. Baseline characteristics were collected and those prescribed dabigatran were followed up for 2 years in Phase II. Phase III, which started when dabigatran had been more widely adopted, gathered data for up to 3 years, regardless of antithrombotic management [8].

Consecutive patients from 38 countries were enrolled between 2014 and 2016. Adult patients with recently diagnosed nonvalvular AF (<3 months before the baseline visit; Latin America <4.5 months) at risk of stroke (CHA$_2$DS$_2$-VASc score ≥1) achieved by any of the following: heart failure or left ventricular systolic dysfunction, hypertension, diabetes, prior stroke, transient ischemic attack (TIA) or systemic embolism, myocardial infarction (MI),

peripheral artery disease, age $\geq$65 years, or female sex, were enrolled [9]. The risks of stroke and bleeding were assessed using the $CHA_2DS_2$-VASc and HAS-BLED (1 point is achieved by any of the following: hypertension, abnormal renal or hepatic function, prior stroke, bleeding or predisposition, labile International Normalised Ratio, elderly [>65 years], or concomitant use of alcohol or anti-inflammatory medications) [10]. Antithrombotic therapy was prescribed by the treating physicians according to local standards. This report is focused on baseline data obtained from patients in Phase III, collected using electronic case report forms.

## Statistical analysis

Baseline characteristics are summarized descriptively. Categorical variables are reported as absolute frequencies and percentages, and continuous variables are summarized by median (Quartile 1, Quartile 3). Baseline characteristics included stratification of patients with AF and multimorbidity according to stroke prevention strategies (OAC vs antiplatelet vs no antithrombotic therapy, NOAC vs vitamin K antagonists [VKAs], and NOACs once daily [QD] vs twice daily [BID]). Standardized differences were used to compare baseline characteristics across various stroke prevention strategies, focusing on variables with the highest standardized differences; differences $\leq$10% in absolute value were considered as balanced between groups [11].

Factors associated with antithrombotic treatment choice were analyzed by log-binomial, multivariable regression models, providing relative probability ratios for prescription (OAC vs no OAC use, NOAC vs VKA; and by region). Missing data were handled using multiple imputation, replacing missing data with multiple simulated values based on regression models to provide comparatively unbiased estimates under the missing-at-random assumption. The procedure introduces random error to compensate for the added, imputed information. The imputation regression models used 56 predictors to impute the missing data, and were repeated 20 times to give 20 datasets with imputed data [12].

Confidence intervals were calculated based on likelihood ratios and Rubin's method to combine results across imputations. Both univariate and multivariable log-binomial regression analyses were performed to evaluate crude as well as the adjusted probability ratios together with 95% confidence intervals. The term "probability ratio" was used rather than "risk ratio", as our measure describes treatment selections rather than adverse outcomes.

All data were calculated using SAS version 9.4 (SAS Institute, Inc., Cary, NC).

## Results

Of 21,241 eligible patients in this subanalysis, 15,119 (71.2%) had $\geq$2 concomitant, chronic conditions (Table 1).

## Baseline characteristics of AF multimorbid patients

Baseline characteristics of patients are summarized based on antithrombotic therapy (Table 2). Among multimorbid AF patients, 83.8% were prescribed OACs, 11.0% were prescribed antiplatelet therapy, and 5.2% were prescribed no antithrombotic therapy. The median (66.0, 79.0) age was 73.0 years in the OAC group, 71.0 (63.0–79.0) years in the antiplatelet therapy group, and 72.0 (64.0–80.0) years in the no antithrombotic therapy group. The proportions of females in these groups were 44.5%, 41.7%, and 45.5%, respectively. The median $CHA_2DS_2$-VASc and HAS-BLED scores were similar across the 3 groups.

Baseline characteristics of patients prescribed NOACs or VKAs are shown in Table 3. The median age was 73.0 (66.0–79.0) years, and the proportion of females was 44% in both treatment groups. There were no differences in $CHA_2DS_2$-VASc and HAS-BLED scores between

**Table 1. Proportion of AF patients according to number of comorbid diseases[a].**

| Number of Comorbid Diseases | Number of Patients (n = 21,241) |
|---|---|
| 0 | 1434 (6.8) |
| 1 | 4688 (22.1) |
| 2 | 5559 (26.2) |
| 3 | 4286 (20.2) |
| 4 | 2664 (12.5) |
| 5 | 1463 (6.9) |
| 6 | 695 (3.3) |
| 7 | 332 (1.6) |
| 8 | 88 (0.4) |
| 9 | 22 (0.1) |
| 10 | 8 (0.0) |
| 11 | 2 (0.0) |

[a]AF = atrial fibrillation.

these 2 groups. The prevalence of paroxysmal AF in patients with multimorbidity on NOACs and VKAs was 57.0% and 45.4%, respectively. Among patients on NOACs, 38.4% had a European Heart Rhythm Association symptom score of I, compared with 33.3% for patients on VKAs. A lower proportion (1.6%) of patients on NOACs had a glomerular filtration rate of 15–29 mL/min, compared with 4.4% of those on VKAs.

Cardioversion was performed in 19.9% of patients on NOACs vs 14.6% of those on VKAs. Treatment in specialist offices was more prevalent for patients on NOACs (33.5% vs 23.8% in the VKA group), while comorbidities such as heart failure (HF) and MI were less prevalent among patients given NOACs.

Patient demographics, cardiovascular risk factors, comorbid diseases, AF categorization, stroke and bleeding risks, and concomitant treatments of patients on NOACs QD vs BID are summarized in Table 4 There were generally small differences between patients taking NOACs QD vs BID. Previous TIA or stroke were present in 14.9% of the patients on NOACs QD vs 21.3% of the patients on NOACs BID (Table 4).

## Factors associated with OAC non-prescription in multimorbid AF patients globally

Results from univariate analyses are presented in the S1 File. In the multivariable log-binomial regression analysis, factors associated with prescriptions for no OAC use in multimorbid AF patients were: type of AF (paroxysmal/persistent vs permanent), coronary artery disease (CAD), MI, history of bleeding, smoking status (current vs nonsmoker), and region (Asia, North America vs Europe). Factors associated with increased OAC use were: age 65–74 vs ≥75 years, body mass index (BMI) class (≥25 vs 18.5–24 kg/m$^2$), creatinine clearance (30–59 vs ≥80 mL/min), hypertension, prior TIA or stroke, and AF ablation (Table 5).

## Factors associated with OACs non-prescription in multimorbid AF patients in Asia, Europe, and North America

Factors associated with prescriptions for no OAC use in multimorbid AF patients in Asia, Europe, and North America are presented in **S1 Table in** S2 File. Factors associated with increased OAC use are included in **S1 Table in** S2 File.

**Table 2. Baseline characteristics of AF multimorbid patients prescribed OAC or antiplatelets or no antithrombotic therapy[a].**

| | OAC (n = 12,677) | Antiplatelets (n = 1658) | No Antithrombotic Therapy (n = 784) |
|---|---|---|---|
| Age (y), median (Q1, Q3) | 73.0 (66.0–79.0) | 71.0 (63.0–79.0) | 72.0 (64.0–80.0) |
| Females, n (%) | 5645 (44.5) | 691 (41.7) | 357 (45.5) |
| BMI (kg/m$^2$), median (Q1, Q3) | 28.0 (24.8–32.0) | 26.1 (23.5–30.0) | 26.1 (23.4–29.6) |
| Missing | 123 (1.0) | 17 (1.0) | 8 (1.0) |
| Current smoker | 1145 (9.0) | 223 (13.4) | 100 (12.8) |
| Alcohol abuse, ≥8 units/ week | 866 (6.8) | 85 (5.1) | 54 (6.9) |
| Type of AF, n (%) | | | |
| Paroxysmal | 6810 (53.7) | 1166 (70.3) | 496 (63.3) |
| Persistent | 4478 (35.3) | 401 (24.2) | 242 (30.9) |
| Permanent | 1389 (11.0) | 91 (5.5) | 46 (5.9) |
| Categorization of AF, n (%) | | | |
| EHRA I | 4686 (37.0) | 550 (33.2) | 273 (34.8) |
| EHRA II | 4025 (31.8) | 563 (34.0) | 270 (34.4) |
| EHRA III | 3063 (24.2) | 431 (26.0) | 183 (23.3) |
| EHRA IV | 903 (7.1) | 114 (6.9) | 58 (7.4) |
| Creatinine clearance (mL/min) (measured), median (Q1, Q3) | 70.6 (52.5–95.3) | 69.5 (50.9–92.4) | 67.8 (49.7–90.3) |
| Creatinine clearance (mL/min), n (%) | | | |
| <15 | 100 (0.8) | 18 (1.1) | 10 (1.3) |
| 15–29 | 305 (2.4) | 62 (3.7) | 23 (2.9) |
| 30–49 | 1848 (14.6) | 252 (15.2) | 136 (17.3) |
| 50–79 | 4152 (32.8) | 526 (31.7) | 253 (32.3) |
| ≥80 | 4080 (32.2) | 520 (31.4) | 243 (31.0) |
| Missing | 2192 (17.3) | 280 (16.9) | 119 (15.2) |
| CHA$_2$DS$_2$-VASc score, median (Q1, Q3) | 4.0 (3.0–5.0) | 4.0 (2.0–5.0) | 3.0 (2.0–4.0) |
| HAS-BLED score, median (Q1, Q3) | 1.0 (1.0–2.0) | 2.0 (2.0–3.0) | 1.0 (1.0–2.0) |
| Missing (HAS-BLED), n (%) | 1234 (9.7) | 134 (8.1) | 69 (8.8) |
| Medical history, n (%) | | | |
| Congestive heart failure | 3509 (27.7) | 487 (29.4) | 215 (27.4) |
| Hypertension | 10,989 (86.7) | 1370 (82.6) | 638 (81.4) |
| Diabetes mellitus | 4021 (31.7) | 510 (30.8) | 226 (28.8) |
| Previous stroke or TIA | 2347 (18.5) | 336 (20.3) | 159 (20.3) |
| Myocardial infarction | 1580 (12.5) | 384 (23.2) | 58 (7.4) |
| Coronary artery disease | 3017 (23.8) | 745 (44.9) | 149 (19.0) |
| Peripheral artery disease | 503 (4.0) | 79 (4.8) | 21 (2.7) |
| Cancer | 1671 (13.2) | 167 (10.1) | 115 (14.7) |
| Dementia | 101 (0.8) | 18 (1.1) | 1 (0.1) |
| Gastric ulcer | 145 (1.1) | 20 (1.2) | 13 (1.7) |
| Gastritis or duodenitis | 455 (3.6) | 70 (4.2) | 50 (6.4) |
| Chronic kidney disease | 3881 (30.6) | 526 (31.7) | 271 (34.6) |
| COPD | 1045 (8.2) | 120 (7.2) | 59 (7.5) |
| Bleeding (after diagnosis of AF), n (%) | 182 (1.4) | 32 (1.9) | 33 (4.2) |
| Bleeding on OAC, n (%) | 159 (87.4) | 27 (84.4) | 18 (54.5) |
| Location of bleeding (after diagnosis of AF), n (%)* | | | |
| Intracranial hemorrhage | 12 (6.6) | 6 (18.8) | 8 (24.2) |
| Upper GI bleed | 12 (6.6) | 4 (12.5) | 3 (9.1) |
| Lower GI bleed | 25 (13.7) | 6 (18.8) | 5 (15.2) |
| GI bleed not further specified | 11 (6.0) | 4 (12.5) | 4 (12.1) |

(*Continued*)

**Table 2.** (Continued)

| | OAC (n = 12,677) | Antiplatelets (n = 1658) | No Antithrombotic Therapy (n = 784) |
|---|---|---|---|
| Urogenital hemorrhage | 31 (17.0) | 3 (9.4) | 3 (9.1) |
| Bleeding at other location | 81 (44.5) | 7 (21.9) | 8 (24.2) |
| Bleeding with unknown location | 10 (5.5) | 2 (6.3) | 2 (6.1) |
| Region, n (%) | | | |
| Asia | 1739 (13.7) | 719 (43.4) | 325 (41.5) |
| Europe | 6514 (51.4) | 443 (26.7) | 266 (33.9) |
| North America | 3429 (27.0) | 415 (25.0) | 144 (18.4) |
| Latin America | 995 (7.8) | 81 (4.9) | 49 (6.3) |
| Type of site, n (%) | | | |
| GP/primary care | 686 (5.4) | 171 (10.3) | 77 (9.8) |
| Specialist office | 3902 (30.8) | 512 (30.9) | 191 (24.4) |
| Community hospital | 3757 (29.6) | 350 (21.1) | 175 (22.3) |
| University hospital | 3878 (30.6) | 543 (32.8) | 326 (41.6) |
| Outpatient health care centre | 222 (1.8) | 51 (3.1) | 6 (0.8) |
| Anticoagulation clinics | 82 (0.6) | 6 (0.4) | 4 (0.5) |
| Other | 150 (1.2) | 25 (1.5) | 5 (0.6) |

[a]AF = atrial fibrillation; BMI = body mass index; CHA$_2$DS$_2$-VASc = congestive heart failure/left ventricular dysfunction, hypertension, age $\geq$75 years, diabetes, stroke/transient ischemic attack/systemic embolism, vascular disease, age 65–74 years, sex category (female); COPD = chronic obstructive pulmonary disease; EHRA = European Heart Rhythm Association; GI = gastrointestinal; GP = general practitioner; HAS-BLED = hypertension, abnormal renal /liver function, stroke, bleeding history or predisposition, labile International Normalised Ratio, elderly (>65 years), drugs or alcohol concomitantly; OAC = oral anticoagulant; Q = quartile; TIA = transient ischemic attack; y = years.

*Proportion calculated out of Bleeding (after diagnosis of AF).

## Factors associated with type of OAC use in multimorbid AF patients globally

Factors associated with prescriptions for VKA use globally in multimorbid AF patients were: age <75 vs $\geq$75 years, MI, congestive HF, diabetes mellitus, creatinine clearance (<60 vs $\geq$80 mL/min), **S2 Table in** S2 File.

Factors associated with decreased VKA use globally were: type of AF (paroxysmal/persistent vs permanent), previous TIA or stroke, medical treatment reimbursement (self-pay/no coverage vs not self-pay), **S2 Table in** S2 File.

## Factors associated with OAC use in multimorbid AF patients in Asia, Europe, North America, and Latin America

Factors associated with prescriptions for VKA use in multimorbid AF patients in Asia, Europe, North America, and Latin America are presented in **S3 Table in** S2 File. Factors associated with decreased prescriptions for VKA use in multimorbid AF patients in Asia, Europe, North America, and Latin America are presented in **S3 Table in** S2 File.

## Discussion

There are still knowledge gaps in how OACs are used in clinical practice in patients with AF and multiple comorbidities and which factors influence OAC prescription in such patients. Our study shows that, despite a median CHA$_2$DS$_2$-VASc score >3, approximately 16% of patients with multimorbidity and AF are not anticoagulated. The baseline characteristics in

**Table 3. Baseline characteristics of AF multimorbid patients prescribed NOACs or VKAs[a].**

| | NOAC (n = 9105) | VKA (n = 3572) | Standardized Difference |
|---|---|---|---|
| Age (y), median (Q1, Q3) | 73.0 (66.0–79.0) | 73.0 (66.0–79.0) | 0.005 |
| Females, n (%) | 4072 (44.7) | 1573 (44.0) | −0.014 |
| BMI (kg/m$^{2)}$, median (Q1, Q3) | 28.0 (24.8–32.2) | 27.8 (24.6–31.6) | −0.066 |
| Missing | 37 (1.2) | 60 (1.0) | 0.020 |
| Current smoker | 812 (8.9) | 333 (9.3) | 0.014 |
| Alcohol abuse, ≥8 units/ week | 651 (7.1) | 215 (6.0) | −0.046 |
| Type of AF, n (%) | | | |
| Paroxysmal | 5187 (57.0) | 1623 (45.4) | −0.232 |
| Persistent | 3052 (33.5) | 1426 (39.9) | 0.133 |
| Permanent | 866 (9.5) | 523 (14.6) | 0.158 |
| Categorization of AF, n (%) | | | |
| EHRA I | 3496 (38.4) | 1190 (33.3) | −0.106 |
| EHRA II | 2886 (31.7) | 1139 (31.9) | 0.004 |
| EHRA III | 2131 (23.4) | 932 (26.1) | 0.062 |
| EHRA IV | 592 (6.5) | 311 (8.7) | 0.083 |
| Creatinine clearance (mL/min) (measured), median (Q1, Q3) | 72.1 (53.7–97.0) | 66.8 (48.9–91.0) | −0.078 |
| Creatinine clearance (mL/min) n (%) | | | |
| <15 | 50 (0.5) | 50 (1.4) | 0.087 |
| 15–29 | 148 (1.6) | 157 (4.4) | 0.163 |
| 30–49 | 1280 (14.1) | 568 (15.9) | 0.052 |
| 50–79 | 3046 (33.5) | 1106 (31.0) | −0.053 |
| ≥80 | 3053 (33.5) | 1027 (28.8) | −0.103 |
| Missing | 1528 (16.8) | 664 (18.6) | 0.047 |
| CHA$_2$DS$_2$-VASc score, median (Q1, Q3) | 4.0 (3.0–5.0) | 4.0 (3.0–5.0) | 0.080 |
| HAS-BLED score, median (Q1, Q3) | 1.0 (1.0–2.0) | 1.0 (1.0–2.0) | 0.016 |
| Missing (HAS-BLED), n (%) | 858 (9.4) | 376 (10.5) | 0.037 |
| Medical history, n (%) | | | |
| Congestive heart failure | 2232 (24.5) | 1277 (35.8) | 0.247 |
| Hypertension | 7907 (86.8) | 3082 (86.3) | −0.016 |
| Diabetes mellitus | 2839 (31.2) | 1182 (33.1) | 0.041 |
| Previous stroke or TIA | 1741 (19.1) | 606 (17.0) | −0.056 |
| Myocardial infarction | 1039 (11.4) | 541 (15.1) | 0.110 |
| Coronary artery disease | 2104 (23.1) | 913 (25.6) | 0.057 |
| Peripheral artery disease | 355 (3.9) | 148 (4.1) | 0.012 |
| Cancer | 1223 (13.4) | 448 (12.5) | −0.027 |
| Dementia | 76 (0.8) | 25 (0.7) | −0.016 |
| Gastric ulcer | 111 (1.2) | 34 (1.0) | −0.026 |
| Gastritis or duodenitis | 317 (3.5) | 138 (3.9) | 0.020 |
| Chronic kidney disease | 2663 (29.2) | 1218 (34.1) | 0.104 |
| COPD | 743 (8.2) | 302 (8.5) | 0.011 |
| Bleeding (after diagnosis of AF), n (%) | 130 (1.4) | 52 (1.5) | 0.002 |
| Bleeding on OAC, n (%) | 112 (86.2) | 47 (90.4) | 0.132 |
| Location of bleeding (after diagnosis of AF), n (%)* | | | |
| Intracranial hemorrhage | 11 (8.5) | 1 (1.9) | −0.298 |
| Upper GI bleed | 8 (6.2) | 4 (7.7) | 0.061 |
| Lower GI bleed | 20 (15.4) | 5 (9.6) | −0.175 |
| GI bleed not further specified | 9 (6.9) | 2 (3.8) | −0.137 |

*(Continued)*

**Table 3.** (Continued)

| | NOAC (n = 9105) | VKA (n = 3572) | Standardized Difference |
|---|---|---|---|
| Urogenital hemorrhage | 20 (15.4) | 11 (21.2) | 0.150 |
| Bleeding at other location | 56 (43.1) | 25 (48.1) | 0.101 |
| Bleeding with unknown location | 6 (4.6) | 4 (7.7) | 0.128 |
| AF cardioversion | 1814 (19.9) | 521 (14.6) | −0.142 |
| Region, n (%) | | | |
| Asia | 1222 (13.4) | 517 (14.5) | 0.030 |
| Europe | 4498 (49.4) | 2016 (56.4) | 0.141 |
| North America | 2808 (30.8) | 621 (17.4) | −0.319 |
| Latin America | 577 (6.3) | 418 (11.7) | 0.188 |
| Type of site, n (%) | | | |
| GP/primary care | 502 (5.5) | 184 (5.2) | −0.016 |
| Specialist office | 3053 (33.5) | 849 (23.8) | −0.217 |
| Community hospital | 2880 (31.6) | 877 (24.6) | −0.158 |
| University hospital | 2454 (27.0) | 1424 (39.9) | 0.276 |
| Outpatient health care centre | 72 (0.8) | 150 (4.2) | 0.220 |
| Anticoagulation clinics | 37 (0.4) | 45 (1.3) | 0.094 |
| Other | 107 (1.2) | 43 (1.2) | 0.003 |

[a]AF = atrial fibrillation; BMI = body mass index; $CHA_2DS_2$-VASc = congestive heart failure/left ventricular dysfunction, hypertension, age ≥75 years, diabetes, stroke/transient ischemic attack/systemic embolism, vascular disease, age 65–74 years, sex category (female); COPD = chronic obstructive pulmonary disease; EHRA = European Heart Rhythm Association; GI = gastrointestinal; GP = general practitioner; HAS-BLED = hypertension, abnormal renal /liver function, stroke, bleeding history or predisposition, labile International Normalised Ratio, elderly (>65 years), drugs or alcohol concomitantly; NOAC = nonvitamin K antagonist oral anticoagulants; OAC = oral anticoagulant; Q = quartile; TIA = transient ischemic attack; VKA = vitamin K antagonists; y = years.
*Proportion calculated out of Bleeding (after diagnosis of AF).

these complex patients differ in relation to antithrombotic therapy selection, suggesting that comorbidities may influence antithrombotic therapy prescription patterns for patients with AF. For example, prescription of OACs globally in patients with AF and multimorbidity was associated with age, BMI, cardiovascular risk factors (smoking status), AF pattern, concomitant diseases (ie, hypertension, CAD, MI, previous TIA or stroke), history of bleeding, renal function, rhythm control strategy (AF ablation and AF cardioversion), and region (Asia and North America). Prescriptions patterns were also subject to regional differences in clinical practice.

## Patient characteristics according to antithrombotic therapy use

The results suggest that patients with AF and multimorbidity prescribed NOACs are more likely to have paroxysmal AF, and have fewer comorbidities than those prescribed VKAs, consistent with other reports [13–15]. Declining renal function may influence the choice of VKA in those with chronic kidney disease. Healthcare system-related factors (such as center type) also influence treatment strategies. Patients with AF and multimorbidity treated in specialist offices and community hospitals are more often prescribed NOACs than VKAs.

The patients in this cohort prescribed antiplatelet agents had a higher risk of bleeding according to HAS-BLED score than those who were prescribed OACs. They also more often had paroxysmal AF compared to those prescribed OACs. Patients with AF and CAD were more often prescribed antiplatelets than OACs despite the fact that antiplatelet therapy does not prevent stroke or reduce mortality, elevates the risk of bleeding, and is not recommended

**Table 4. Baseline characteristics of AF multimorbid patients prescribed NOACs QD or NOACs BID.**

| | NOAC QD (n = 3071) | NOAC BID (n = 6034) | Standardized Difference |
|---|---|---|---|
| Age (y), median (Q1, Q3) | 72.0 (65.0–79.0) | 73.0 (66.0–79.0) | −0.098 |
| Females, n (%) | 1306 (42.5) | 2766 (45.8) | −0.067 |
| BMI (kg/m$^2$), median (Q1, Q3) | 28.3 (25.0–32.8) | 27.9 (24.8–32.0) | 0.089 |
| Current smoker | 250 (8.1) | 562 (9.3) | −0.042 |
| Alcohol abuse, ≥8 units/ week | 242 (7.9) | 409 (6.8) | 0.042 |
| Type of AF, n (%) | | | |
| Paroxysmal | 1767 (57.5) | 3420 (56.7) | 0.017 |
| Persistent | 1045 (34.0) | 2007 (33.3) | 0.016 |
| Permanent | 259 (8.4) | 607 (10.1) | −0.056 |
| Categorization of AF, n (%) | | | |
| EHRA I | 1138 (37.1) | 2358 (39.1) | −0.042 |
| EHRA II | 983 (32.0) | 1903 (31.5) | 0.010 |
| EHRA III | 775 (25.2) | 1356 (22.5) | 0.065 |
| EHRA IV | 175 (5.7) | 417 (6.9) | −0.050 |
| Creatinine clearance (mL/min), (measured), median (Q1, Q3) | 74.4 (55.3–101.8) | 70.5 (53.1–94.3) | 0.041 |
| Creatinine clearance, n (%) | | | |
| <15 | 18 (0.6) | 32 (0.5) | 0.008 |
| 15–29 | 40 (1.3) | 108 (1.8) | −0.040 |
| 30–49 | 401 (13.1) | 879 (14.6) | −0.044 |
| 50–79 | 1018 (33.1) | 2028 (33.6) | −0.010 |
| ≥80 | 1125 (36.6) | 1928 (32.0) | 0.099 |
| Missing | 469 (15.3) | 1059 (17.6) | −0.062 |
| CHA$_2$DS$_2$-VASc score, median (Q1, Q3) | 3.0 (2.0–4.0) | 4.0 (3.0–5.0) | −0.127 |
| HAS-BLED score, median (Q1, Q3) | 1.0 (1.0–2.0) | 1.0 (1.0–2.0) | −0.066 |
| Missing (HAS-BLED), n (%) | 302 (9.8) | 556 (9.2) | 0.021 |
| Medical history, n (%) | | | |
| Congestive heart failure | 772 (25.1) | 1460 (24.2) | 0.022 |
| Hypertension | 2672 (87.0) | 5235 (86.8) | 0.007 |
| Diabetes mellitus | 1021 (33.2) | 1818 (30.1) | 0.067 |
| Previous stroke or TIA | 457 (14.9) | 1284 (21.3) | −0.167 |
| Myocardial infarction | 366 (11.9) | 673 (11.2) | 0.024 |
| Coronary artery disease | 746 (24.3) | 1358 (22.5) | 0.042 |
| Peripheral artery disease | 119 (3.9) | 236 (3.9) | −0.002 |
| Cancer | 407 (13.3) | 816 (13.5) | −0.008 |
| Dementia | 24 (0.8) | 52 (0.9) | −0.009 |
| Gastric ulcer | 40 (1.3) | 71 (1.2) | 0.011 |
| Gastritis or duodenitis | 116 (3.8) | 201 (3.3) | 0.024 |
| Chronic kidney disease | 839 (27.3) | 1824 (30.2) | −0.064 |
| COPD | 258 (8.4) | 485 (8.0) | 0.013 |
| Bleeding (after diagnosis of AF), n (%) | 57 (1.9) | 73 (1.2) | 0.053 |
| Bleeding on OAC, n (%) | 52 (91.2) | 60 (82.2) | 0.269 |
| Location of bleeding (after diagnosis of AF), n (%) | | | |
| Intracranial hemorrhage | 2 (3.5) | 9 (12.3) | −0.331 |
| Upper GI bleed | 4 (7.0) | 4 (5.5) | 0.064 |
| Lower GI bleed | 10 (17.5) | 10 (13.7) | 0.106 |
| GI bleed not further specified | 5 (8.8) | 4 (5.5) | 0.128 |
| Urogenital hemorrhage | 6 (10.5) | 14 (19.2) | −0.245 |

*(Continued)*

**Table 4.** (Continued)

| | NOAC QD (n = 3071) | NOAC BID (n = 6034) | Standardized Difference |
|---|---|---|---|
| Bleeding at other location | 24 (42.1) | 32 (43.8) | −0.035 |
| Bleeding with unknown location | 6 (10.5) | 0 (0.0) | 0.438 |
| AF cardioversion | 710 (23.1) | 1104 (18.3) | 0.119 |
| Region, n (%) | | | |
| Asia | 356 (11.6) | 866 (14.4) | −0.082 |
| Europe | 1465 (47.7) | 3033 (50.3) | −0.051 |
| North America | 1056 (34.4) | 1752 (29.0) | 0.115 |
| Latin America | 194 (6.3) | 383 (6.3) | −0.001 |
| Type of site, n (%) | | | |
| GP/primary care | 184 (6.0) | 318 (5.3) | 0.031 |
| Specialist office | 1110 (36.1) | 1943 (32.2) | 0.083 |
| Community hospital | 921 (30.0) | 1959 (32.5) | −0.053 |
| University hospital | 773 (25.2) | 1681 (27.9) | −0.061 |
| Outpatient health care center | 19 (0.6) | 53 (0.9) | −0.030 |
| Anticoagulation clinics | 18 (0.6) | 19 (0.3) | 0.041 |
| Other | 46 (1.5) | 61 (1.0) | 0.044 |

[a]AF = atrial fibrillation; BID = twice daily; BMI = body mass index; $CHA_2DS_2$-VASc = congestive heart failure/left ventricular dysfunction, hypertension, age ≥75 years, diabetes, stroke/transient ischemic attack/systemic embolism, vascular disease, age 65–74 years, sex category (female); COPD = chronic obstructive pulmonary disease; EHRA = European Heart Rhythm Association; GI = gastrointestinal; GP = general practitioner; HAS-BLED = hypertension, abnormal renal /liver function, stroke, bleeding history or predisposition, labile International Normalised Ratio, elderly (>65 years), drugs or alcohol concomitantly; NOAC = nonvitamin K antagonist oral anticoagulants; OAC = oral anticoagulant; Q = quartile; QD = once daily; TIA = transient ischemic attack.

for prevention of AF-related thromboembolism [16]. Unfortunately, antiplatelet monotherapy is still a frequent choice of prescribing physicians based on several European reports [17, 18].

## Factors associated with OAC prescription in multimorbid AF patients globally

The majority of multimorbid AF patients had a high risk of stroke ($CHA_2DS_2$-VASc score ≥2) and oral anticoagulation therapy is recommended for these patients [19]. Hypertension and HF were the most prevalent risk factors for thromboembolic complications [20] and these factors and previous stroke or TIA are associated with a greater frequency of OAC prescription. Prescription of OACs was inversely associated with comorbidities that are strongly associated with elevated thromboembolic risk (eg, MI, CAD), just as conditions associated with an increased risk of bleeding (eg, previous hemorrhagic events) were associated with less frequent prescription of OACs. This is also consistent with prior reports [13] although current clinical practice guidelines recommend that patients with AF at a high risk of bleeding should generally continue anticoagulation with frequent visits and close monitoring [21]. A history of AF ablation in multimorbid AF patients was associated with more frequent OAC prescription as per guidelines [21] and consistent with other studies [22].

Younger age (≤75 years) was associated with greater OAC prescription and more frequent selection of VKAs compared to practice patterns for older patients. Several studies have suggested that increasing age is a barrier to implementing OAC use [23, 24]. Importantly, stroke risk increases with age, and the absolute benefit of OACs is clearly increased for older patients with AF [25]. In one report, when adjusted for comorbidity, age was not an important determinant of anticoagulation [26].

**Table 5. Multivariable log-binomial analysis for factors associated with prescription of OAC therapy (no OAC vs OAC)[a,b].**

| Factor | Relative Risk (95% CI) For Prescription of No OAC Globally |
|---|---|
| Age | |
| <65 | 1.05 (0.95–1.16) |
| 65–74 | 0.90 (0.83–0.99) |
| ≥75 | 1.0 (ref) |
| BMI class | |
| <18.5 | 0.98 (0.77–1.24) |
| 18.5–24 | 1.0 (ref) |
| 25–29 | 0.85 (0.79–0.91) |
| 30–34 | 0.77 (0.69–0.87) |
| ≥35 | 0.70 (0.60–0.81) |
| Gender | |
| Male | 1.0 (ref) |
| Female | 1.05 (0.97–1.13) |
| Current smoker | 1.14 (1.03–1.25) |
| Past smoker | 0.91 (0.84–0.99) |
| Categorization of AF | |
| EHRA I | 1.0 (ref) |
| EHRA II | 1.04 (0.96–1.12) |
| EHRA III | 0.99 (0.91–1.07) |
| EHRA IV | 1.07 (0.95–1.20) |
| Type of AF | |
| Paroxysmal | 1.67 (1.42–1.97) |
| Persistent | 1.20 (1.02–1.43) |
| Permanent | 1.0 (ref) |
| Hypertension | 0.89 (0.83–0.97) |
| Coronary artery disease | 1.42 (1.31–1.53) |
| Myocardial infarction | 1.18 (1.08–1.28) |
| Congestive heart failure | 1.01 (0.94–1.08) |
| Diabetes mellitus | 0.95 (0.88–1.02) |
| Previous TIA or stroke | 0.81 (0.68–0.97) |
| Bleeding after diagnosis of AF | 1.60 (1.42–1.79) |
| Peripheral artery disease | 1.13 (0.96–1.34) |
| Cancer | 1.00 (0.90–1.12) |
| Functional dyspepsia | 0.85 (0.56–1.27) |
| Gastric ulcer | 0.91 (0.69–1.21) |
| Gastritis or duodenitis | 0.95 (0.82–1.10) |
| COPD | 1.03 (0.90–1.19) |
| Hyperthyroidism | 0.96 (0.79–1.17) |
| Hepatic disease | 1.05 (0.87–1.27) |
| Dementia | 1.09 (0.76–1.56) |
| AF cardioversion | 0.96 (0.89–1.04) |
| Creatinine clearance (mL/min) | |
| <30 | 1.09 (0.94–1.26) |
| 30–59 | 0.88 (0.79–0.97) |
| 60–79 | 0.91 (0.83–1.00) |
| ≥80 | 1.0 (ref) |

*(Continued)*

**Table 5.** (Continued)

| Factor | Relative Risk (95% CI) For Prescription of No OAC Globally |
|---|---|
| AF ablation | 0.30 (0.20–0.45) |
| Region | |
| Asia | 3.17 (2.88–3.49) |
| Europe | 1.0 (ref) |
| North America | 1.24 (1.11–1.39) |
| Latin America | 1.14 (0.96–1.37) |
| Medical treatment reimbursed by | |
| Self-pay/no coverage | 0.82 (0.69–0.96) |
| Not self-pay | 1.0 (ref) |
| Type of site | |
| Specialist office | 1.26 (1.14–1.39) |
| Community hospital | 1.0 (ref) |
| University hospital | 1.28 (1.17–1.40) |

[a] A few other variables (alcohol abuse, psychosocial factors, biological heart valve implant, valve repair, and peptic ulcer) are included in the multivariable log-binomial regression analysis model and are presented in the S1 File.
[b] AF = atrial fibrillation; BMI = body mass index; CI = confidence interval; COPD = chronic obstructive pulmonary disease; EHRA = European Heart Rhythm Association; OAC = oral anticoagulant; ref = reference; TIA = transient ischemic attack.

Multimorbid AF patients with paroxysmal or persistent AF were less often prescribed OACs in particular VKAs than those with permanent AF. NOACs should be preferred in patients with multimorbidity and polypharmacy given their lower number of drug–drug interactions compared with VKAs [27]. Ischemic stroke may occur as frequently in paroxysmal AF as in permanent AF, especially with multiple risk factors [28]. Moreover, the use of OACs should be based on stroke risk assessment according to the $CHA_2DS_2$-VASc risk score [21]. The pattern of AF seems to be related to patient profiles characterized by age, concomitant diseases, symptoms, and risk factors for stroke and bleeding [13]. Patients with higher European Heart Rhythm Association symptom scores were more often prescribed VKAs than those who were asymptomatic.

Multimorbid AF patients with a history of cardioversion were less often prescribed VKAs than those without prior cardioversion. NOACs were preferred in multimorbid AF patients after cardioversion. A similar pattern was found in another study where rhythm control strategy was associated with selection of NOAC [14].

### OAC prescription in multimorbid AF patients regionally

In this study, multimorbidity influenced ATT use within particular regions. In Europe, younger patients (age <65 years) were less likely to be prescribed OACs than older patients (age ≥75 years). Multimorbid AF patients with congestive HF were more likely to be anticoagulated due to an increased risk of thromboembolism. In Europe, bleeding risk of a patient as perceived by physicians may be the reason for decreased use of anticoagulation. Patients with gastritis or duodenitis or hepatic disease are less likely to be prescribed OACs, probably because of the elevated risk of bleeding. This association has been previously noted [26]. In Asia, younger patients (age <75 years) were more likely to be prescribed OACs than older patients (age ≥75 years). Interestingly, patients with gastritis or duodenitis or a history of cancer were more likely to receive OAC than those without those diseases. In North America,

younger multimorbid AF patients (age <65 years) were less likely to be prescribed OACs than older patients (age ≥75 years). Multimorbid AF patients with diabetes were more likely to receive OACs, due to their association with higher thromboembolic risk, as well as higher all-cause, cardiovascular, and noncardiovascular mortality [29]. AF patients with multimorbidity and cancer in North America were less likely to receive OAC.

Asia and North America were associated with decreased OAC prescription. In Asia, OACs are less commonly prescribed in nonvalvular AF patients than in Europe, possibly because of suspicion of the risk of bleeding during treatment [30]. Also, NOACs are not reimbursed in some Asian countries.

## Strengths

It is one the largest prospective global cohort of consecutive AF patients receiving different antithrombotic treatments. Initiation of Phase III was region-specific, once relevant baseline characteristics of patients initiating dabigatran and VKA therapy in Phase II overlapped based on propensity score comparisons. After the baseline visit, all patients in this Phase III were managed according to local clinical practice and were followed for 3 years, regardless of pre-scribed antithrombotic therapy. This study had regular follow-up with physicians, alongside on-site monitoring, multiple standards for data quality assurance and review.

## Limitations

Although the GLORIA-AF study was designed to capture all outcome events, this analysis did not consider follow-up data. The following limitations exist in our study: we have no data on patient and prescriber treatment preferences; similarly, reasons for OAC nonprescription were not reported. Furthermore, this study reflects single, initial-treatment decisions during a period when prescribing patterns may have been changing, and the analysis was based on pre-scription pattern shortly after entry into the registry (baseline). Neither have we accounted for quality of anticoagulation or changes in clinical practice patterns over time.

## Conclusion

AF patients with multimorbidity who were prescribed NOACs were relatively healthier, more likely to have paroxysmal AF, and had fewer prevalent comorbidities than AF multimorbid patients on VKAs. Multimorbidity may determine the antithrombotic therapy prescription pattern within AF patients. Several factors are related to increased OAC prescription in multi-morbid AF patients, including younger age, hypertension, prior TIA or stroke, and AF abla-tion. Pattern of AF (paroxysmal and persistent AF), CAD, MI, history of bleeding, and region (Asia, North America) were inversely associated with OAC prescription.

## Supporting information

**S1 File.**
(PDF)

**S2 File.**
(ODT)

## Acknowledgments

The authors thank the patients who participated in this trial, their families, the investigators, study co-ordinators, and study teams.

GLORIA_AF Phase III Participating Investigator listing.

Dzifa Wosornu Abban

Nasser Abdul

Atilio Marcelo Abud

Fran Adams

Srinivas Addala

Pedro Adragão

Walter Ageno

Rajesh Aggarwal

Sergio Agosti

Piergiuseppe Agostoni

Francisco Aguilar

Julio Aguilar Linares

Luis Aguinaga

Jameel Ahmed

Allessandro Aiello

Paul Ainsworth

Jorge Roberto Aiub

Raed Al-Dallow

Lisa Alderson

Jorge Antonio Aldrete Velasco

Dimitrios Alexopoulos

Fernando Alfonso Manterola

Pareed Aliyar

David Alonso

Fernando Augusto Alves da Costa

José Amado

Walid Amara

Mathieu Amelot

Nima Amjadi

Fabrizio Ammirati

Marianna Andrade

Nabil Andrawis

Giorgio Annoni

Gerardo Ansalone

M.Kevin Ariani

Juan Carlos Arias

Sébastien Armero

Chander Arora

Muhammad Shakil Aslam

M. Asselman

Philippe Audouin

Charles Augenbraun

S. Aydin

Ivaneta Ayryanova

Emad Aziz

Luciano Marcelo Backes

E. Badings

Ermentina Bagni

Seth H. Baker
Richard Bala
Antonio Baldi
Shigenobu Bando
Subhash Banerjee
Alan Bank
Gonzalo Barón Esquivias
Craig Barr
Maria Bartlett
Vanja Basic Kes
Giovanni Baula
Steffen Behrens
Alan Bell
Raffaella Benedetti
Juan Benezet Mazuecos
Bouziane Benhalima
Jutta Bergler-Klein
Jean-Baptiste Berneau
Richard A. Bernstein
Percy Berrospi
Sergio Berti
Andrea Berz
Elizabeth Best
Paulo Bettencourt
Robert Betzu
Ravi Bhagwat
Luna Bhatta
Francesco Biscione
Giovanni BISIGNANI
Toby Black
Michael J. Bloch
Stephen Bloom
Edwin Blumberg
Mario Bo
Ellen Bøhmer
Andreas Bollmann
Maria Grazia Bongiorni
Giuseppe Boriani
D.J. Boswijk
Jochen Bott
Edo Bottacchi
Marica Bracic Kalan
Drew Bradman
Donald Brautigam
Nicolas Breton
P.J.A.M. Brouwers
Kevin Browne
Jordi Bruguera Cortada
A. Bruni

Claude Brunschwig
Hervé Buathier
Aurélie Buhl
John Bullinga
Jose Walter Cabrera
Alberto Caccavo
Shanglang Cai
Sarah Caine
Leonardo Calò
Valeria Calvi
Mauricio Camarillo Sánchez
Rui Candeias
Vincenzo Capuano
Alessandro Capucci
Ronald Caputo
Tatiana Cárdenas Rizo
Francisco Cardona
Francisco Carlos da Costa Darrieux
Yan Carlos Duarte Vera
Antonio Carolei
Susana Carreño
Paula Carvalho
Susanna Cary
Gavino Casu
Claudio Cavallini
Guillaume Cayla
Aldo Celentano
Tae-Joon Cha
Kwang Soo Cha
Jei Keon Chae
Kathrine Chalamidas
Krishnan Challappa
Sunil Prakash Chand
Harinath Chandrashekar
Ludovic Chartier
Kausik Chatterjee
Carlos Antero Chavez Ayala
Aamir Cheema
Amjad Cheema
Lin Chen
Shih-Ann Chen
Jyh Hong Chen
Fu-Tien Chiang
Francesco Chiarella
Lin Chih-Chan
Yong Keun Cho
Jong-Il Choi
Dong Ju Choi
Guy Chouinard

Danny Hoi-Fan Chow
Dimitrios Chrysos
Galina Chumakova
Eduardo Julián José Roberto Chuquiure Valenzuela
Nicoleta Cindea Nica
David J. Cislowski
Anthony Clay
Piers Clifford
Andrew Cohen
Michael Cohen
Serge Cohen
Furio Colivicchi
Ronan Collins
Paolo Colonna
Steve Compton
Derek Connolly
Alberto Conti
Gabriel Contreras Buenostro
Gregg Coodley
Martin Cooper
Julian Coronel
Giovanni Corso
Juan Cosín Sales
Yves Cottin
John Covalesky
Aurel Cracan
Filippo Crea
Peter Crean
James Crenshaw
Tina Cullen
Harald Darius
Patrick Dary
Olivier Dascotte
Ira Dauber
Vicente Davalos
Ruth Davies
Gershan Davis
Jean-Marc Davy
Mark Dayer
Marzia De Biasio
Silvana De Bonis
Raffaele De Caterina
Teresiano De Franceschi
J.R. de Groot
José De Horta
Axel De La Briolle
Gilberto de la Pena Topete
Angelo Amato Vicenzo de Paola
Weimar de Souza

A. de Veer
Luc De Wolf
Eric Decoulx
Sasalu Deepak
Pascal Defaye
Freddy Del-Carpio Munoz
Diana Delic Brkljacic
N. Joseph Deumite
Silvia Di Legge
Igor Diemberger
Denise Dietz
Pedro Dionísio
Qiang Dong
Fabio Rossi dos Santos
Elena Dotcheva
Rami Doukky
Anthony D'Souza
Simon Dubrey
Xavier Ducrocq
Dmitry Dupljakov
Mauricio Duque
Dipankar Dutta
Nathalie Duvilla
A. Duygun
Rainer Dziewas
Charles B. Eaton
William Eaves
L.A Ebels-Tuinbeek
Clifford Ehrlich
Sabine Eichinger-Hasenauer
Steven J. Eisenberg
Adnan El Jabali
Mahfouz El Shahawy
Mauro Esteves Hernandes
Ana Etxeberria Izal
Rudolph Evonich III
Oksana Evseeva
Andrey Ezhov
Raed Fahmy
Quan Fang
Ramin Farsad
Laurent Fauchier
Stefano Favale
Maxime Fayard
Jose Luis Fedele
Francesco Fedele
Olga Fedorishina
Steven R. Fera
Luis Gustavo Gomes Ferreira

Jorge Ferreira
Claudio Ferri
Anna Ferrier
Hugo Ferro
Alexandra Finsen
Brian First
Stuart Fischer
Catarina Fonseca
Luísa Fonseca Almeida
Steven Forman
Brad Frandsen
William French
Keith Friedman
Athena Friese
Ana Gabriela Fruntelata
Shigeru Fujii
Stefano Fumagalli
Marta Fundamenski
Yutaka Furukawa
Matthias Gabelmann
Nashwa Gabra
Niels Gadsbøll
Michel Galinier
Anders Gammelgaard
Priya Ganeshkumar
Christopher Gans
Antonio Garcia Quintana
Olivier Gartenlaub
Achille Gaspardone
Conrad Genz
Frédéric Georger
Jean-Louis Georges
Steven Georgeson
Evaldas Giedrimas
Mariusz Gierba
Ignacio Gil Ortega
Eve Gillespie
Alberto Giniger
Michael C. Giudici
Alexandros Gkotsis
Taya V. Glotzer
Joachim Gmehling
Jacek Gniot
Peter Goethals
Seth Goldbarg
Ronald Goldberg
Britta Goldmann
Sergey Golitsyn
Silvia Gómez

Juan Gomez Mesa
Vicente Bertomeu Gonzalez
Jesus Antonio Gonzalez Hermosillo
Víctor Manuel González López
Hervé Gorka
Charles Gornick
Diana Gorog
Venkat Gottipaty
Pascal Goube
Ioannis Goudevenos
Brett Graham
G. Stephen Greer
Uwe Gremmler
Paul G. Grena
Martin Grond
Edoardo Gronda
Gerian Grönefeld
Xiang Gu
Ivett Guadalupe Torres Torres
Gabriele Guardigli
Carolina Guevara
Alexandre Guignier
Michele Gulizia
Michael Gumbley
Albrecht Günther
Andrew Ha
Georgios Hahalis
Joseph Hakas
Christian Hall
Bing Han
Seongwook Han
Joe Hargrove
David Hargroves
Kenneth B. Harris
Tetsuya Haruna
Emil Hayek
Jeff Healey
Steven Hearne
Michael Heffernan
Geir Heggelund
J.A. Heijmeriks
Maarten Hemels
I. Hendriks
Sam Henein
Sung-Ho Her
Paul Hermany
Jorge Eduardo Hernández Del Río
Yorihiko Higashino
Michael Hill

Tetsuo Hisadome
Eiji Hishida
Etienne Hoffer
Matthew Hoghton
Kui Hong
Suk keun Hong
Stevie Horbach
Masataka Horiuchi
Yinglong Hou
Jeff Hsing
Chi-Hung Huang
David Huckins
kathy Hughes
A. Huizinga
E.L. Hulsman
Kuo-Chun Hung
Gyo-Seung Hwang
Margaret Ikpoh
Davide Imberti
Hüseyin Ince
Ciro Indolfi
Shujiro Inoue
Didier Irles
Harukazu Iseki
C. Noah Israel
Bruce Iteld
Venkat Iyer
Ewart Jackson-Voyzey
Naseem Jaffrani
Frank Jäger
Martin James
Sung-Won Jang
Nicolas Jaramillo
Nabil Jarmukli
Robert J. Jeanfreau
Ronald D. Jenkins
Carlos Jerjes Sánchez
Javier Jimenez
Robert Jobe
Tomas Joen-Jakobsen
Nicholas Jones
Jose Carlos Moura Jorge
Bernard Jouve
Byung Chun Jung
Kyung Tae Jung
Werner Jung
Mikhail Kachkovskiy
Krystallenia Kafkala
Larisa Kalinina

Bernd Kallmünzer
Farzan Kamali
Takehiro Kamo
Priit Kampus
Hisham Kashou
Andreas Kastrup
Apostolos Katsivas
Elizabeth Kaufman
Kazuya Kawai
Kenji Kawajiri
John F. Kazmierski
P Keeling
José Francisco Kerr Saraiva
Galina Ketova
AJIT Singh Khaira
Aleksey Khripun
Doo-Il Kim
Young Hoon Kim
Nam Ho Kim
Dae Kyeong Kim
Jeong Su Kim
June Soo Kim
Ki Seok Kim
Jin bae Kim
Elena Kinova
Alexander Klein
James J. Kmetzo
G. Larsen Kneller
Aleksandar Knezevic
Su Mei Angela Koh
Shunichi Koide
Athanasios Kollias
J.A. Kooistra
Jay Koons
Martin Koschutnik
William J. Kostis
Dragan Kovacic
Jacek Kowalczyk
Natalya Koziolova
Peter Kraft
Johannes A. Kragten
Mori Krantz
Lars Krause
B.J. Krenning
F. Krikke
Z. Kromhout
Waldemar Krysiak
Priya Kumar
Thomas Kümler

Malte Kuniss
Jen-Yuan Kuo
Achim Küppers
Karla Kurrelmeyer
Choong Hwan Kwak
Bénédicte Laboulle
Arthur Labovitz
Wen Ter Lai
Andy Lam
Yat Yin Lam
Fernando Lanas Zanetti
Charles Landau
Giancarlo Landini
Estêvão Lanna Figueiredo
Torben Larsen
Karine Lavandier
Jessica LeBlanc
Moon Hyoung Lee
Chang-Hoon Lee
John Lehman
Ana Leitão
Nicolas Lellouche
Malgorzata Lelonek
Radoslaw Lenarczyk
T. Lenderink
Salvador León González
Peter Leong-Sit
Matthias Leschke
Nicolas Ley
Zhanquan Li
Xiaodong Li
Weihua Li
Xiaoming Li
Christhoh Lichy
Ira Lieber
Ramon Horacio Limon Rodriguez
Hailong Lin
Gregory Y. H. Lip
Feng Liu
Hengliang Liu
Guillermo Llamas Esperon
Nassip Llerena Navarro
Eric Lo
Sergiy Lokshyn
Amador López
José Luís López-Sendón
Adalberto Menezes Lorga Filho
Richard S. Lorraine
Carlos Alberto Luengas

Robert Luke
Ming Luo
Steven Lupovitch
Philippe Lyrer
Changsheng Ma
Genshan Ma
Irene Madariaga
Koji Maeno
Dominique Magnin
Gustavo Maid
Sumeet K. Mainigi
Konstantinos Makaritsis
Rohit Malhotra
Rickey Manning
Athanasios Manolis
Helard Andres Manrique Hurtado
Ioannis Mantas
Fernando Manzur Jattin
Vicky Maqueda
Niccolo Marchionni
Francisco Marin Ortuno
Antonio Martín Santana
Jorge Martinez
Petra Maskova
Norberto Matadamas Hernandez
Katsuhiro Matsuda
Tillmann Maurer
Ciro Mauro
Erik May
Nolan Mayer
John McClure
Terry McCormack
William McGarity
Hugh McIntyre
Brent McLaurin
Feliz Alvaro Medina Palomino
Francesco Melandri
Hiroshi Meno
Dhananjai Menzies
Marco Mercader
Christian Meyer
Beat j. Meyer
Jacek Miarka
Frank Mibach
Dominik Michalski
Patrik Michel
Rami Mihail Chreih
Ghiath Mikdadi
Milan Mikus

Davor Milicic
Constantin Militaru
Sedi Minaie
Bogdan Minescu
Iveta Mintale
Tristan Mirault
Michael J. Mirro
Dinesh Mistry
Nicoleta Violeta Miu
Naomasa Miyamoto
Tiziano Moccetti
Akber Mohammed
Azlisham Mohd Nor
Michael Mollerus
Giulio Molon
Sergio Mondillo
Patrícia Moniz
Lluis Mont
Vicente Montagud
Oscar Montaña
Cristina Monti
Luciano Moretti
Kiyoo Mori
Andrew Moriarty
Jacek Morka
Luigi Moschini
Nikitas Moschos
Andreas Mügge
Thomas J. Mulhearn
Carmen Muresan
Michela Muriago
Wlodzimierz Musial
Carl W. Musser
Francesco Musumeci
Thuraia Nageh
Hidemitsu Nakagawa
Yuichiro Nakamura
Toru Nakayama
Gi-Byoung Nam
Michele Nanna
Indira Natarajan
Hemal M. Nayak
Stefan Naydenov
Jurica Nazlić
Alexandru Cristian Nechita
Libor Nechvatal
Sandra Adela Negron
James Neiman
Fernando Carvalho Neuenschwander

David Neves
Anna Neykova
Ricardo Nicolás Miguel
George Nijmeh
Alexey Nizov
Rodrigo Noronha Campos
Janko Nossan
Tatiana Novikova
Ewa Nowalany-Kozielska
Emmanuel Nsah
Juan Carlos Nunez Fragoso
Svetlana Nurgalieva
Dieter Nuyens
Ole Nyvad
Manuel Odin de Los Rios Ibarra
Philip O'Donnell
Martin O'Donnell
Seil Oh
Yong Seog Oh
Dongjin Oh
Gilles O'Hara
Kostas Oikonomou
Claudia Olivares
Richard Oliver
Rafael Olvera Ruiz
Christoforos Olympios
Anna omaszuk-Kazberuk
Joaquín Osca Asensi
eena Padayattil jose
Francisco Gerardo Padilla Padilla
Victoria Padilla Rios
Giuseppe Pajes
A. Shekhar Pandey
Gaetano Paparella
F Paris
Hyung Wook Park
Jong Sung Park
Fragkiskos Parthenakis
Enrico Passamonti
Rajesh J. Patel
Jaydutt Patel
Mehool Patel
Janice Patrick
Ricardo Pavón Jimenez
Analía Paz
Vittorio Pengo
William Pentz
Beatriz Pérez
Alma Minerva Pérez Ríos

Alejandro Pérez-Cabezas
Richard Perlman
Viktor Persic
Francesco Perticone
Terri K. Peters
Sanjiv Petkar
Luis Felipe Pezo
Christian Pflücke
David N. Pham
Roland T. Phillips
Stephen Phlaum
Denis Pieters
Julien Pineau
Arnold Pinter
Fausto Pinto
R. Pisters
Nediljko Pivac
Darko Pocanic
Cristian Podoleanu
Alessandro Politano
Zdravka Poljakovic
Stewart Pollock
Jose Polo Garcéa
Holger Poppert
Maurizio Porcu
Antonio Pose Reino
Neeraj Prasad
Dalton Bertolim Précoma
Alessandro Prelle
John Prodafikas
Konstantin Protasov
Maurice Pye
Zhaohui Qiu
Jean-Michel Quedillac
Dimitar Raev
Carlos Antonio Raffo Grado
Sidiqullah Rahimi
Arturo Raisaro
Bhola Rama
Ricardo Ramos
Maria Ranieri
Nuno Raposo
Eric Rashba
Ursula Rauch-Kroehnert
Ramakota Reddy
Giulia Renda
Shabbir Reza
Luigi Ria
Dimitrios Richter

Hans Rickli
Werner Rieker
Tomas Ripolil Vera
Luiz Eduardo Ritt
Douglas Roberts
Ignacio Rodriguez Briones
Aldo Edwin Rodriguez Escudero
Carlos Rodríguez Pascual
Mark Roman
Francesco Romeo
E. Ronner
Jean-Francois Roux
Nadezda Rozkova
Miroslav Rubacek
Frank Rubalcava
Andrea M. Russo
Matthieu Pierre Rutgers
Karin Rybak
Samir Said
Tamotsu Sakamoto
Abraham Salacata
Adrien Salem
Rafael Salguero Bodes
Marco A. Saltzman
Alessandro Salvioni
Gregorio Sanchez Vallejo
Marcelo Sanmartín Fernández
Wladmir Faustino Saporito
Kesari Sarikonda
Taishi Sasaoka
Hamdi Sati
Irina Savelieva
Pierre-Jean Scala
Peter Schellinger
Carlos Scherr
Lisa Schmitz
Karl-Heinz Schmitz
Bettina Schmitz
Teresa Schnabel
Steffen Schnupp
Peter Schoeniger
Norbert Schön
Peter Schwimmbeck
Clare Seamark
Greg Searles
Karl-Heinz Seidl
Barry Seidman
Jaroslaw Sek
Lakshmanan Sekaran

Carlo Serrati
Neerav Shah
Vinay Shah
Anil Shah
Shujahat Shah
Vijay Kumar Sharma
Louise Shaw
Khalid H. Sheikh
Naruhito Shimizu
Hideki Shimomura
Dong-Gu Shin
Eun-Seok Shin
Junya Shite
Gerolamo Sibilio
Frank Silver
Iveta Sime
Tim A. Simmers
Narendra Singh
Peter Siostrzonek
Didier Smadja
David W. Smith
Marcelo Snitman
Dario Sobral Filho
Hassan Soda
Carl Sofley
Adam Sokal
Yannie Soo Oi Yan
Rodolfo Sotolongo
Olga Ferreira de Souza
Jon Arne Sparby
Jindrich Spinar
David Sprigings
Alex C. Spyropoulos
Dimitrios Stakos
Clemens Steinwender
Georgios Stergiou
Ian Stiell
Marcus Stoddard
Anastas Stoikov
Witold Streb
Ioannis Styliadis
Guohai Su
Xi Su
Wanda Sudnik
Kai Sukles
Xiaofei Sun
H. Swart
Janko Szavits-Nossan
Jens Taggeselle

Yuichiro Takagi
Amrit Pal Singh Takhar
Angelika Tamm
Katsumi Tanaka
Tanyanan Tanawuttiwat
Sherman Tang
Aylmer Tang
Giovanni Tarsi
Tiziana Tassinari
Ashis Tayal
Muzahir Tayebjee
J.M. ten Berg
Dan Tesloianu
Salem H.K. The
Dierk Thomas
Serge Timsit
Tetsuya Tobaru
Andrzej R. Tomasik.
Mikhail Torosoff
Emmanuel Touze
Elina Trendafilova
W. Kevin Tsai
Hung Fat Tse
Hiroshi Tsutsui
Tian Ming Tu
Ype Tuininga
Minang Turakhia
Samir Turk
Wayne Turner
Arnljot Tveit
Richard Tytus
C Valadão
P.F.M.M. van Bergen
Philippe van de Borne
B.J. van den Berg
C van der Zwaan
M. Van Eck
Peter Vanacker
Dimo Vasilev
Vasileios Vasilikos
Maxim Vasilyev
Srikar Veerareddy
Mario Vega Miño
Asok Venkataraman
Paolo Verdecchia
Francesco Versaci
Ernst Günter Vester
Hubert Vial
Jason Victory

Alejandro Villamil
Marc Vincent
Anthony Vlastaris
Jürgen vom Dahl
Kishor Vora
Robert B. Vranian
Paul Wakefield
Ningfu Wang
Mingsheng Wang
Xinhua Wang
Feng Wang
Tian Wang
Alberta L. Warner
Kouki Watanabe
Jeanne Wei
Christian Weimar
Stanislav Weiner
Renate Weinrich
Ming-Shien Wen
Marcus Wiemer
Preben Wiggers
Andreas Wilke
David Williams
Marcus L. Williams
Bernhard Witzenbichler
Brian Wong
Ka Sing Lawrence Wong
Beata Wozakowska-Kaplon
Shulin Wu
Richard C. Wu
Silke Wunderlich
Nell Wyatt
John (Jack) Wylie
Yong Xu
Xiangdong Xu
Hiroki Yamanoue
Takeshi Yamashita
Ping Yen Bryan Yan
Tianlun Yang
Jing Yao
Kuo-Ho Yeh
Wei Hsian Yin
Yoto Yotov
Ralf Zahn
Stuart Zarich
Sergei Zenin
Elisabeth Louise Zeuthen
Huanyi Zhang
Donghui Zhang

Xingwei Zhang

Ping Zhang

Jun Zhang

Shui Ping Zhao

Yujie Zhao

Zhichen Zhao

Yang Zheng

Jing Zhou

Sergio Zimmermann

Andrea Zini

Steven Zizzo

Wenxia Zong

L Steven Zukerman

## Author Contributions

**Conceptualization:** Monika Kozieł, Christine Teutsch, Jonathan L. Halperin, Kenneth J. Rothman, Hans-Christoph Diener, Chang-Sheng Ma, Sabrina Marler, Shihai Lu, Menno V. Huisman, Gregory Y. H. Lip.

**Formal analysis:** Sabrina Marler, Shihai Lu, Gregory Y. H. Lip.

**Investigation:** Christine Teutsch, Sabrina Marler, Shihai Lu, Venkatesh K. Gurusamy, Gregory Y. H. Lip.

**Methodology:** Monika Kozieł, Christine Teutsch, Jonathan L. Halperin, Kenneth J. Rothman, Sabrina Marler, Shihai Lu, Gregory Y. H. Lip.

**Resources:** Christine Teutsch, Venkatesh K. Gurusamy.

**Software:** Shihai Lu, Venkatesh K. Gurusamy.

**Supervision:** Hans-Christoph Diener, Menno V. Huisman, Gregory Y. H. Lip.

**Validation:** Jonathan L. Halperin, Kenneth J. Rothman, Gregory Y. H. Lip.

**Writing – original draft:** Monika Kozieł.

**Writing – review & editing:** Christine Teutsch, Jonathan L. Halperin, Kenneth J. Rothman, Hans-Christoph Diener, Chang-Sheng Ma, Sabrina Marler, Shihai Lu, Venkatesh K. Gurusamy, Menno V. Huisman, Gregory Y. H. Lip.

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
