## [Decision Letter · Decision Letter 0]

25 Jan 2021

PONE-D-20-40256

Manuscript Type: Original article

Atrial Fibrillation and Comorbidities: Clinical Characteristics and Antithrombotic Treatment in GLORIA-AF

PLOS ONE

Dear Dr. Kozieł,

Thank you for submitting your manuscript to PLOS ONE. After careful consideration, we feel that it has merit but does not fully meet PLOS ONE’s publication criteria as it currently stands. Therefore, we invite you to submit a revised version of the manuscript that addresses the points raised during the review process.

The study has been revised by two academics, with expertise in the field of the study and its statatistical methods. The methodology employed is partly correct and sound in general, as assessed by the reviewers. However, there are some specific aspects of the statistical methods used that should be revised/clarified. One point is the type of treatment that the individuals are employing. If the treatment changes, the authors should indicate how they account for this change in their model. A reorganization of the manuscript, to a certain extent, is suggested by one reviewer in the attached document, aiming at a more understandable presentation of the data. Further, the reviewer recommends including a further discussion/comparison with recent data on different cohorts.

We look forward to receiving your revised manuscript.

Kind regards,

Pablo Garcia de Frutos

Academic Editor

PLOS ONE

Journal Requirements:

2. Thank you for submitting your clinical trial to PLOS ONE and for providing the name of the registry and the registration number. The information in the registry entry suggests that your trial was registered after patient recruitment began. PLOS ONE strongly encourages authors to register all trials before recruiting the first participant in a study.

1) your reasons for your delay in registering this study (after enrolment of participants started);

2) confirmation that all related trials are registered by stating: “The authors confirm that all ongoing and related trials for this drug/intervention are registered”.

"Dr Kozieł and Professor Rothman declare no conflicts of interest.

Dr Teutsch, Dr Lu, Sabrina Marler, and Venkatesh K. Gurusamy are employees of Boehringer Ingelheim.

Professor Halperin has engaged in consulting activities for Boehringer Ingelheim and advisory activities involving anticoagulants, and he is a member of the Executive Steering Committee of the GLORIA-AF Registry.

Over the past 3 years, Professor Diener received honoraria for participation in clinical trials, contribution to advisory boards, or oral presentations from: Abbott, Bayer Vital, Bristol-Myers Squibb, Boehringer Ingelheim, Daiichi Sankyo, Medtronic, Pfizer, Portola, Sanofi-Aventis, and WebMD Global. Financial support for research projects was provided by Boehringer Ingelheim. He received research grants from the German Research Council (DFG), German Ministry of Education and Research (BMBF), European Union, NIH, Bertelsmann Foundation, and Heinz-Nixdorf Foundation.

Professor Ma received honoraria from Bristol-Myers Squibb, Pfizer, Johnson & Johnson, Boehringer Ingelheim, Bayer, and AstraZeneca for giving lectures.

Professor Huisman reports grants from ZonMW Dutch Healthcare Fund, grants and personal fees from Boehringer Ingelheim, Pfizer/Bristol-Myers Squibb, Bayer Health Care, Aspen, Daiichi Sankyo, outside the submitted work.

Professor Lip has been a consultant for Bayer/Janssen, Bristol-Myers Squibb/Pfizer, Medtronic, Boehringer Ingelheim, Novartis, Verseon, and Daiichi Sankyo. He has been a speaker for Bayer, Bristol-Myers Squibb/Pfizer, Medtronic, Boehringer Ingelheim, and Daiichi Sankyo. No fees directly received personally."

We note that one or more of the authors have an affiliation to the commercial funders of this research study : Boehringer Ingelheim. We also note that one or more of the authors are employed by a commercial company: RTI Health Solutions.

3.1. Please provide an amended Funding Statement declaring this commercial affiliation, as well as a statement regarding the Role of Funders in your study. If the funding organization did not play a role in the study design, data collection and analysis, decision to publish, or preparation of the manuscript and only provided financial support in the form of authors' salaries and/or research materials, please review your statements relating to the author contributions, and ensure you have specifically and accurately indicated the role(s) that these authors had in your study. You can update author roles in the Author Contributions section of the online submission form.

3.2. Please also provide an updated Competing Interests Statement declaring this commercial affiliation along with any other relevant declarations relating to employment, consultancy, patents, products in development, or marketed products, etc.  

5. Please amend either the title on the online submission form (via Edit Submission) or the title in the manuscript so that they are identical.

6. One of the noted authors is a group or consortium [GLORIA-AF Investigators]. In addition to naming the author group, please list the individual authors and affiliations within this group in the acknowledgments section of your manuscript. Please also indicate clearly a lead author for this group along with a contact email address.

Reviewers' comments:

Reviewer's Responses to Questions

**Comments to the Author**

1. Is the manuscript technically sound, and do the data support the conclusions?

Reviewer #1: Yes

Reviewer #2: Partly

2. Has the statistical analysis been performed appropriately and rigorously? 

Reviewer #1: Yes

Reviewer #2: No

3. Have the authors made all data underlying the findings in their manuscript fully available?

Reviewer #1: Yes

Reviewer #2: No

4. Is the manuscript presented in an intelligible fashion and written in standard English?

Reviewer #1: Yes

Reviewer #2: Yes

5. Review Comments to the Author

Reviewer #1: The authors of the paper “Atrial Fibrillation and Comorbidities: Clinical Characteristics and Antithrombotic Treatment in GLORIA-AF" have been focused on the analysis objective, specifically in baseline characteristics and antithrombotic therapy in patients with AF and more than two concomitant and chronic comorbidities.

For more information, please check the attached file.

Reviewer #2: PONE-D-20-40256: statistical review

SUMMARY. This study describes factors that are associated with oral anticoagulant (OAC) prescription in subjects with atrial fibrillation and comorbidities. I am not sure that the association between patient's characteristics and prescription makes sense as a research question. However, I'm not a medical doctor and I am going to limit my discussion to the statistical methods that are deployed here.

From a statistical viewpoint, the core of the analysis relies on a cross-sectional log-binomial regression model, estimated by likelihood-based multiple imputation methods. These methods could in principle be appropriate, but the paper lacks information about the data structure and the multiple imputation method. I therefore need to ask first some questions (major issues 1 and 2 below) before making a recommendation.

MAJOR ISSUES

1. Little is said about the structure of the response variable: OAC prescriptions. The cross-sectional log-binomial regression model assumes that the response variable is a binary variable. It therefore makes sense if each subject receive only one type of prescription during the whole follow up. In this case, the model chosen provides a correct approach. If instead subjects switch between a type of prescription to another one, then this method is no longer correct and the data must be examined by a longitudinal version of the model that includes subject-specific random effects. Please clarify.

2. The authors correctly use multiple imputation to handle missing data in a regression framework. However, nothing is said about the model used to generate the imputation sample. If all the incomplete covariates are continuous, a multivariate normal distribution is typically used for imputation. However, in this study, incomplete covariates are of mixed type (some are continuous, others are categorical). I therefore wonder what model has been used at the imputation step.

6. PLOS authors have the option to publish the peer review history of their article (what does this mean?). If published, this will include your full peer review and any attached files.

Reviewer #1: No

Reviewer #2: No

---

## [Author Response · Author response to Decision Letter 0]

4 Mar 2021

Reviewer #1

The authors of the paper “Atrial Fibrillation and Comorbidities: Clinical Characteristics and Antithrombotic Treatment in GLORIA-AF" have been focused on the analysis objective, specifically in baseline characteristics and antithrombotic therapy in patients with AF and more than two concomitant and chronic comorbidities. Authors have done, through Gloria-AF registry, an international covered registry, a genuine work in screening of baseline characteristics and medication of an enormous number of patients with AF during a three-year enrolling period. Even though we have the baseline picture, it is of a big interest to know the trend or changes with regards to oral anticoagulation (OAC) therapy during the follow-up (FU), but as mentioned in Limitations the authors have chosen to show the FU data probably later on. General comments: The manuscript is well written with the objective and methodology clearly described. The authors make an important point showing that patients with AF and a high-risk profile for stroke, more than 2 chronic comorbidities, were unfortunately undertreated with NOAC, in a period when at least three NOACs were established in the clinical use. According to the authors, around 60% of those patients were on NOAC regimen, but still 16% of those patients were without OAC, despite CHA2DS2-VASc ≥ 2. This work points the fact that younger patients with AF and less comorbidities were likely to were on NOAC regimen, at the same time that observations and clinical practice today show something different. In this perspective, I think it is very important the future information from the registry if the trends during the follow-up will show the impact of implementation of the current guidelines for the antithrombotic therapy in patients with AF. I have some comments I would like to address to the authors: 

1. I will appreciate if the authors can explain or argue what was the reason for choosing of standardized differences in order to compare baseline characteristics between different stroke prevention strategies.

Authors’ response:

Thank you. Standardized differences were used because they are independent of sample size and it allows for the comparison of the relative balance of variables measured in different units. 

 2. Again, when considering the trend aspect, it is of interest if a trend of increased NOAC use was observed during the last year of inclusion as compared with the previous year(s).

Authors’ response:

Thank you. Trends of NOAC use during the last years of inclusion as compared with previous years has been addressed by a different manuscript which has been under separate development for submission. 

 3. In Discussion, I feel sometimes that authors give some more results rather than discuss the findings and face them with the literature, and specifically in the second paragraph.

Authors’ response:

Thank you. The Discussion has been edited and the findings have been discussed along with other studies (see pages 8-12).

4. I miss the comparison and discussion with some more recent findings regarding to NOAC use, where data from Scandinavia has given a different picture than probably other regions in Europe. 

Authors’ response:

Thank you. Our findings have been compared and discussed with Scandinavian registries regarding NOAC (see pages 11 and 12). 

‘Of note, younger patients (18-54 years) and ≥75 years were less likely to receive NOAC than those aged 65-74 years in Sweden (31). In Denmark, older age was associated with increased NOAC use (32).’

‘In contrast, HF was associated with decreased OAC initiation in Danish dataset (33).’

‘In Danish nationwide registries, bleeding was also associated with decreased OAC use (33).’

‘In our registry, previous TIA or stroke was the comorbidity associated with decreased VKA use in multimorbid AF patients in Europe. Interestingly, stroke/ thromboembolism or bleeding were associated with increased NOAC initiation in Denmark (32)’.

5. The tables are informative and presented well, but sometimes the perception of overflow is not avoidable. The possibility to cut some information could be considered. 

Authors’ response:

Thank you. The tables have been carefully revised and some excessive information has been deleted (see Tables 2-5). 

6. I would like to have in the ordinary tables one of those included in supplements, specifically table S3 and would recommend authors to change with another one. I believe that one of strengths with this analysis is the region-specific patients enrolling and comparisons.

Authors’ response:

Thank you. The Table S3 have been changed into separate tables for Asia, Europe, North America and South America (see Table S3, S4, S5 and S6). 

Reviewer #2: PONE-D-20-40256: statistical review

SUMMARY. This study describes factors that are associated with oral anticoagulant (OAC) prescription in subjects with atrial fibrillation and comorbidities. I am not sure that the association between patient's characteristics and prescription makes sense as a research question. However, I'm not a medical doctor and I am going to limit my discussion to the statistical methods that are deployed here.

From a statistical viewpoint, the core of the analysis relies on a cross-sectional log-binomial regression model, estimated by likelihood-based multiple imputation methods. These methods could in principle be appropriate, but the paper lacks information about the data structure and the multiple imputation method. I therefore need to ask first some questions (major issues 1 and 2 below) before making a recommendation.

MAJOR ISSUES

1. Little is said about the structure of the response variable: OAC prescriptions. The cross-sectional log-binomial regression model assumes that the response variable is a binary variable. It therefore makes sense if each subject receive only one type of prescription during the whole follow up. In this case, the model chosen provides a correct approach. If instead subjects switch between a type of prescription to another one, then this method is no longer correct and the data must be examined by a longitudinal version of the model that includes subject-specific random effects. Please clarify.

Authors’ response:

Thank you. In this registry, there was no intervention in treatment prescription for patients over time, so patients can have more than one types of treatment prescribed during the study. The cross-sectional association analysis is based on the first prescription of antithrombotic treatment (i.e. index treatment) that was prescribed as long term use at baseline visit. Per inclusion criteria in protocol, the patients enrolled in the study must have been newly diagnosed (<3 months prior to baseline visit) with non-valvular AF. In order to clarify this definition of response variable in the manuscript, we have changed ‘OAC prescription’ to ‘baseline OAC prescription’. 

2. The authors correctly use multiple imputation to handle missing data in a regression framework. However, nothing is said about the model used to generate the imputation sample. If all the incomplete covariates are continuous, a multivariate normal distribution is typically used for imputation. However, in this study, incomplete covariates are of mixed type (some are continuous, others are categorical). I therefore wonder what model has been used at the imputation step.

Authors’ response:

Thank you. The technique called multiple imputation by chained equations or fully conditional specification) was used to impute missing continuous and categorical variables. This method was well described in the reference ‘White IR, Royston P, Wood AM. Multiple imputation using chained equations: issues and guidance for practice. Stat Med. 2011;30(4):377-399.’. It provides the flexibility of handling different types of variables at the same time by employing different types of models accordingly. For example, ordinal linear regression is used for imputing continuous variable; logistic regression or discriminant function is used for (ordinal or nominal) categorical variables. For each of the total 56 covariates analysed, there is one imputation model specified. For this clinically orientated manuscript, the full details of this statistical analysis were not provided. However, we have added one sentence in the ‘statistical analysis’ section to explain what specific multiple imputation is used that can deal with both continuous and categorical variables (see page 6).

‘Multiple imputation by chained equations was used to impute both missing categorical and continuous values.’

---

## [Decision Letter · Decision Letter 1]

22 Mar 2021

Manuscript Type: Original article

Atrial Fibrillation and Comorbidities: Clinical Characteristics and Antithrombotic Treatment in GLORIA-AF

PONE-D-20-40256R1

Dear Dr. Kozieł,

We’re pleased to inform you that your manuscript has been judged scientifically suitable for publication and will be formally accepted for publication once it meets all outstanding technical requirements.

Kind regards,

Pablo Garcia de Frutos

Academic Editor

PLOS ONE

Additional Editor Comments (optional):

Reviewers' comments:

Reviewer's Responses to Questions

**Comments to the Author**

1. If the authors have adequately addressed your comments raised in a previous round of review and you feel that this manuscript is now acceptable for publication, you may indicate that here to bypass the “Comments to the Author” section, enter your conflict of interest statement in the “Confidential to Editor” section, and submit your "Accept" recommendation.

Reviewer #1: All comments have been addressed

Reviewer #2: All comments have been addressed

2. Is the manuscript technically sound, and do the data support the conclusions?

Reviewer #1: Yes

Reviewer #2: (No Response)

3. Has the statistical analysis been performed appropriately and rigorously? 

Reviewer #1: N/A

Reviewer #2: (No Response)

4. Have the authors made all data underlying the findings in their manuscript fully available?

Reviewer #1: Yes

Reviewer #2: (No Response)

5. Is the manuscript presented in an intelligible fashion and written in standard English?

Reviewer #1: Yes

Reviewer #2: (No Response)

6. Review Comments to the Author

Reviewer #1: The authors have answered to my questions and suggestions, and I am satisfied with their comments. They have also improved the discussion and tables.

Reviewer #2: (No Response)

7. PLOS authors have the option to publish the peer review history of their article (what does this mean?). If published, this will include your full peer review and any attached files.

Reviewer #1: No

Reviewer #2: No

---

## [Editor Report · Acceptance letter]

5 Apr 2021

PONE-D-20-40256R1 

Atrial Fibrillation and Comorbidities: Clinical Characteristics and Antithrombotic Treatment in GLORIA-AF 

Dear Dr. Kozieł:

I'm pleased to inform you that your manuscript has been deemed suitable for publication in PLOS ONE. Congratulations! Your manuscript is now with our production department. 

Kind regards, 

on behalf of

Dr. Pablo Garcia de Frutos 

Academic Editor

PLOS ONE